

# Charge quantization and detector resolution

**Roman-Pascal Riwar⋆**

Peter Grünberg Institute, Theoretical Nanoelectronics, Forschungszentrum Jülich,
D-52425 Jülich, Germany

⋆ r.riwar@fz-juelich.de

## Abstract

Charge quantization, or the absence thereof, is a central theme in quantum circuit theory, with dramatic consequences for the predicted circuit dynamics. Very recently, the question of whether or not charge should actually be described as quantized has enjoyed renewed widespread interest, with however seemingly contradictory propositions. Here, we intend to reconcile these different approaches, by arguing that ultimately, charge quantization is not an intrinsic system property, but instead depends on the spatial resolution of the charge detector. We show that the latter can be directly probed by unique geometric signatures in the correlations of the supercurrent. We illustrate these findings at the example of Josephson junction arrays in the superinductor regime, where the transported charge appears to be continuous. Finally, we comment on potential consequences of charge quantization beyond superconducting circuits.

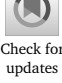

# 1 Introduction

Condensed matter physics is rife with quantum phase transitions where the fundamental notion of charge quantization (in units of the elementary charge $e$) seems challenged, for instance in anyonic field theories [1] with the fractional quantum Hall effect as a famous example [2–4], but also in Luttinger liquids [5–7]. In the language of quantum circuit theory, the charge-phase quantization condition, $[\varphi, N] = i$, stipulates that if the charge $2eN$ [1] is quantized (continuous) then the phase $\varphi$ is compact (noncompact) [8]. For instance, while a regular Josephson junction (JJ) transports integer Cooper pairs ($\varphi$-space is $2\pi$-periodic), junctions involving Majorana- or parafermions, give rise to a $4\pi$ or even $8\pi$ fractional Josephson effect [9–11]. However, also seemingly innocuous circuit elements such as linear inductors appear to break charge quantization (with in fact a continuous quasicharge) [12].

The question of whether or not charge should actually appear as quantized in circuit theory received renewed widespread interest, with two seemingly contradictory propositions, which have radical consequences on the predictions of the circuit dynamics. Inductively shunting a JJ was predicted [12] to suppress any residual sensitivity on the offset charge noise [13], an idea which is at the heart of the fluxonium qubit [14, 15]. There remained a conundrum: charge quantization seemed broken even in the limit of large inductance, when the transport is dominated by the JJ [12]. Recently, a resolution was proposed by advocating continuous charge and noncompact phase irrespective of the presence or absence of a linear inductance [16]. At the same time, there emerged an opposite school of thought in several different contexts, where charge quantization is preserved. A recent theoretical work reevaluated the charge-noise sensitivity of fluxonium qubits [17]. Moreover, the existence of dissipative quantum phase transitions in JJs [18–22] was put into question [23] when assuming a capacitive coupling to the electromagnetic environment (preserving charge quantization), instead of the standard inductive coupling [24]. Finally, charge quantization is important for novel *transport* topological phase transitions, where the topological invariants are defined on a compact $\varphi$-manifold [25–36].

To summarize, the existing literature appears ambiguous. We here aim to build a bridge between the theories of quantized [17,23] versus continuous charge [12,16,37] by developing and illustrating the following two statements (see also Fig. 1):

(i) Charge quantization depends fundamentally on the spatial resolution with which charge is detected or interacted with; it is therefore a property of the measurement basis.

(ii) Current-current correlations provide unique signatures of the detector resolution, in the form of a geometric response which resembles a Zak-Berry phase defined in $\varphi$-space.

---

[1]The prefactor $2e$ expresses that usually, the charge in quantum circuits is counted in units of Cooper pairs.

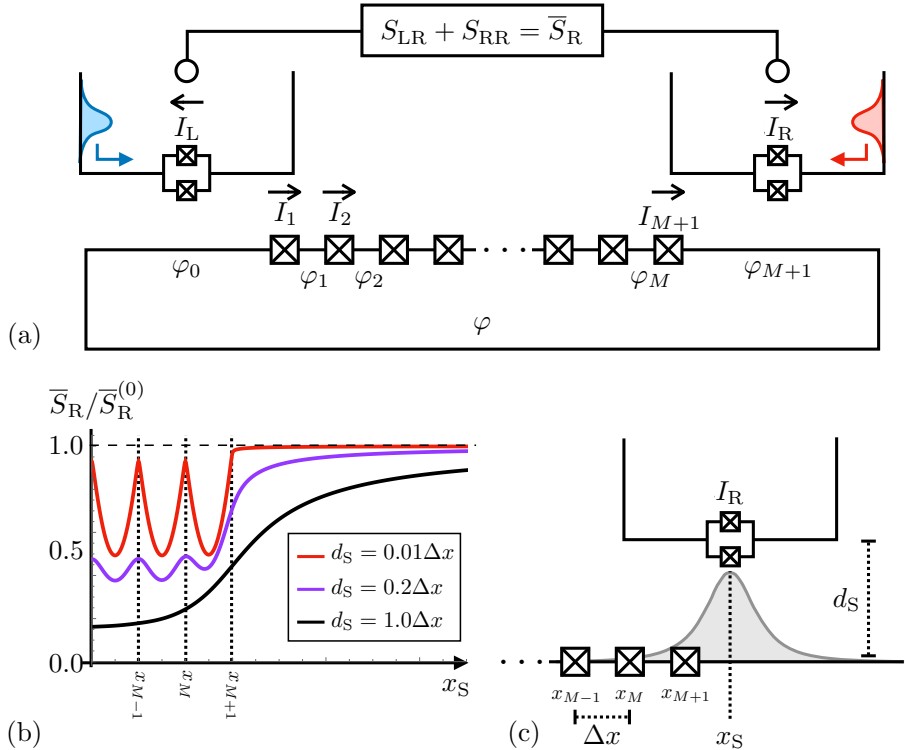

Figure 1: Illustration of the main points. (a) Two SQUID detectors measure the currents to the left ($I_L$) and the right ($I_R$) in a JJ array with a total phase bias $\varphi$, through incoming wave packets (red and blue). The signal of interest is the sum of the current auto and cross-correlations, $\overline{S}_\gamma = \sum_\alpha S_{\alpha\gamma}$ (here, $\gamma = R$). (b) The signal $\overline{S}_R$ depends on the detector properties (c): the lateral position $x_S$ and distance to the circuit $d_S$, determining the detector fuzziness (represented as the grey bell curve). It reaches an extremal value $\overline{S}_R^{(0)}$ when the detector measures the current at a sharp interface. Here, charge is measured in integer units of $2e$. For any $\overline{S}_\gamma/\overline{S}_\gamma^{(0)} < 1$, the detector is fuzzy and fails to resolve integer charges.

Statement (i) implies that in order to determine the correct circuit theory, it has to be carefully analyzed how exactly different parts of the circuit couple to each others charges. We will also comment on potential caveats, respectively, extensions of (i) for topological superconductors. Statement (ii) aims at finding stringent, unique experimental evidence to support (i). We stress that (ii) is in so far highly unexpected, as current-current correlations are low cumulants, which were up to now assumed to be insensitive to charge quantization, respectively to the detector resolution (see discussions for the full-counting statistics of Luttinger liquids [7] and sequential electron tunneling [38]). Furthermore, we expect (see outlook) that an appropriate generalization of above statements beyond quantum circuit theory could open up new research directions investigating the importance of charge quantization in other highly relevant domains, such as in Luttinger liquid theory itself, where charge quantization is discarded in the course of the low-energy approximation [5].

We illustrate the above statements at the example of a JJ array. Apart from their importance as a superinductor element in the fluxonium [14, 15], such arrays are the ideal theoretical testbed. They have a rich existing theoretical literature, which is still actively being expanded [39–42], with a precise circuit model where charges are quantized, and a low-energy regime (formally resembling a Luttinger liquid [40, 42]), where charge appears continuous. Moreover, there has been significant recent experimental progress on JJ arrays, such as mea-

surements illuminating the role of disorder [43], massively increasing chain length, $\sim 10^4$ [44], and alternative pathways for their fabrication, e.g., through granular aluminum [45].

In the following, we first formulate statements (i) and (ii) independent of any model specifics. We then illustrate them at the example of the JJ array. Finally, we comment on important further-reaching consequences of our work, leading to various ideas for follow-up work.

## 2 Quantized charges in circuit theory

### 2.1 Importance of detector resolution

Consider an electronic quantum system, described by a field theory with the electron field operator $\Psi^{(\dagger)}(x)$ [2], satisfying fermionic anti-commutation relations $\{\Psi(x), \Psi^\dagger(x')\} = \delta(x - x')$. It can be easily shown that any local charge number operator $N_e$, defined in an interval $a < x < b$,

$$N_e = \int_a^b dx\, \Psi^\dagger(x)\Psi(x) \tag{1}$$

must have integer eigenvalues. For this purpose, we construct a general basis of charge eigenstates with $n \in \mathbb{N}$ charges in the interval $a < x < b$,

$$
\begin{aligned}
|n, \{k_1, k_2, \ldots, k_n\}\rangle = &\int_a^b dx_1 \int_a^b dx_2 \ldots \int_a^b dx_n\, \psi_{k_1}(x_1)\psi_{k_2}(x_2)\ldots\psi_{k_n}(x_n) \\
&\times \Psi^\dagger(x_1)\Psi^\dagger(x_2)\ldots\Psi^\dagger(x_n)|0\rangle,
\end{aligned}
\tag{2}
$$

where $|0\rangle$ is the absolute vacuum state, $\Psi(x)|0\rangle = 0$ for all $x$. Of course, the functions $\psi_{k_i}$ have to fulfill all the necessary conditions (orthogonality, completeness [3]), which we however do not require explicitly for our proof. Applying the above state to $N_e$, one can show that

$$N_e |n, \{k_1, k_2, \ldots, k_n\}\rangle = n |n, \{k_1, k_2, \ldots, k_n\}\rangle, \tag{3}$$

by simply using fermionic anti-commutation relations for the field $\Psi^{(\dagger)}(x)$. Namely, the annihilation operator $\Psi$ stemming from $N_e$ can be anti-commuted through the chain of $n$ creation operators $\Psi^\dagger$ in $|n\rangle$ for $n$ times, until it destroys the vacuum state $\Psi|0\rangle = 0$, giving rise to the prefactor $n$ in Eq. (3).

As stated in the introduction, there is an abundance of effective field theories in condensed matter physics, where the above seems to no longer apply, due to low-energy approximations of $\Psi$. We stress however, that for any electronic realization of an exotic phase of matter, there must be an underlying microscopic theory in terms of *bare electrons*. Hence, as long as a certain external system or a detector couples to the local electron charge $eN_e$, it will interact with or measure it in integer units, irrespective of any phase transitions. In this sense, any noninteger (e.g., fractional) charge is an *effective* charge (see also a similar argument put forth in Ref. [38]).

---

[2] For simplicity, but without loss of generality, we ignore spin, and consider only one spatial dimension. The statement layed out in the main text can however easily be generalized to include spin and higher spatial dimensions.

[3] Note that here, we focus on constructing a complete many-body basis within the interval from $a$ to $b$. In order to construct a complete many-body basis within the entire system, we need to extend the basis to include adding $n'$ charges to the regions outside this interval. This can be accomplished by a tensor product $|n, n'\rangle = |n\rangle \otimes |n'\rangle$, where $|n'\rangle$ can be constructed in analogy to $|n\rangle$ except with $x \leq a$ or $x \geq b$. Since $N_e$ only acts on $a < x < b$, we still find $N_e|n, n'\rangle = n|n, n'\rangle$.

Crucially, charge quantization as introduced above requires the assumption of a spatially sharp measurement. To illustrate this, let us generalize $N_e$ to

$$N_e[S] = \int dx S(x) \Psi^\dagger(x) \Psi(x), \tag{4}$$

where $S$ is a support function. The charge in Eq. (1) can be obtained from Eq. (4) by assuming sharp boundaries, $S(x) = \theta(x-a)\theta(b-x)$, where $\theta$ is the Heaviside theta function. However, as soon as we allow for $S$ to assume non-integer values, it follows that $N_e[S]$ is no longer guaranteed to have integer eigenvalues. In our work, such a situation will be reached, because of a fuzzy detector, which in general fails to resolve charges with absolute spatial precision (see Fig. 1c).

In the here considered context of conventional superconducting quantum circuits, coherent transport events come in units of Cooper pairs (with charge $2e$), where the Cooper pair number $N = N_e/2$ has a canonically conjugate phase $\varphi$, such that $[\varphi, N] = i$. Consequently, $N$ having integer or continuous eigenvalues can be equivalently encoded in representing $\varphi$ on a compact ($2\pi$-periodic) or non-compact manifold, respectively [8]. Therefore, the presence or absence of charge quantization manifests on the level of the circuit Hamiltonian $H(\varphi)$, in that it is either $2\pi$-periodic, $H(\varphi + 2\pi) = H(\varphi)$, or not. Importantly, note that on this general level, the pair of $\varphi$ and $N$ may refer either to the charge and phase in a (finite) region, *or* to the charge and phase difference across an interface. In quantum circuit theory, these two definitions are commonly referred to as node and branch variables, respectively [46]. The reason for this flexibility is that charge quantization in a given region and quantization of the charge transport into (or out of) this region via a given interface must obviously go hand in hand; the $2\pi$-periodicity constraint on the Hamiltonian must therefore hold irrespective of the use of branch or node variables.

The above leads us to our first observation, statement (i), which has been used in Refs. [17, 23] for particular systems, but which has, to the best of our knowledge, not yet been formulated in this general manner: since charge quantization is fundamentally a property of the detector basis, we conjecture that there must always exist a unitary basis transformation, whereby a Hamiltonian describing the transfer of noninteger charges across an interface, or equivalently, noninteger charges inside a region, can be adequately "requantized". Namely, if a certain theory provides a Hamiltonian $H(\varphi + 2\pi) \neq H(\varphi)$, there must exist a transformation $U(\varphi)$, $\widetilde{H}(\varphi) = U(\varphi)H(\varphi)U^\dagger(\varphi)$, such that $\widetilde{H}(\varphi + 2\pi) = \widetilde{H}(\varphi)$. The transformation $U(\varphi)$ thus connects different choices for the detector basis. In the following, we refer to these choices as gauge choices. Statement (i), as formulated above, will be explicitly illustrated for the JJ array below.

Before proceeding, let us put the above statement into a broader perspective, in particular when considering topological superconductors. Majorana-based Josephson junctions famously give rise to a $4\pi$-Josephson effect. Let us consider as an example the circuit proposed by Fu and Kane [9], which essentially realizes a loop of a Kitaev chain by tunnel-coupling the ends, enclosing a phase $\varphi$ due to an external magnetic flux. This circuit is described by a Hamiltonian of the form,

$$H_{\text{FK}}(\varphi) = 2E_M \cos\left(\frac{\varphi}{2}\right)\left(d_M^\dagger d_M - \frac{1}{2}\right), \tag{5}$$

where $E_M$ is the energy associated to the tunneling of Majorana fermions, and $d_M^{(\dagger)}$ is the operator linking the even and odd parity ground state of the superconducting chain. As such, $H_{\text{FK}}$ is $4\pi$-periodic, representing the fact that the coherent transport via Majorana fermions carries charge $e$ instead of the $2e$ Cooper pair charge of conventional JJs. Note that due to $d_M^\dagger d_M$ having two eigenvalues $0, 1$ (representing the even and odd fermion parity ground state of the circuit), the Hamiltonian has the eigenspectrum $\pm E_M \cos(\varphi/2)$. Therefore, due

to $\cos[(\varphi + 2\pi)/2] = -\cos(\varphi/2)$, it would be possible – at least from a purely mathematical point of view – to find a transformation $U(\varphi)$ to render $H_{\text{FK}}$ $2\pi$-periodic. From a physical point of view however, such a transformation would be problematic, as such a $U(\varphi)$ would give rise to a basis with superpositions of even and odd fermion parity states, thus violating the fermion-parity superselection rule. After all, the breaking of Cooper pairs with charge $2e$ into Majorana-based transport with charge $e$ can be considered physical; since Cooper pairs are obviously composite particles, they can be physically split.

Importantly however, this splitting cannot go any further. The charge $e/2$ associated to the $8\pi$-Josephson effect involving parafermions, as discussed in Refs. [10,11], must ultimately be considered an effective charge, since the electron cannot be physically subdivided into smaller portions, see our general field theoretic argument above. Hence, while it is possible to find an $8\pi$-periodic Hamiltonian describing a parafermion-based JJ [10], we should always find a transformation $U(\varphi)$ to render the Hamiltonian at least $4\pi$-periodic, without running into issues related to parity superselection.

At any rate, in the following, we will focus on conventional superconductors where transport occurs with regular Cooper pairs, such that $2\pi$-periodicity must be obtainable via a basis transformation. For conventional superconductors, only transport processes involving Bogoliubov quasiparticles can change the charge within a given region by $e$ instead of $2e$. Such processes are however dissipative in nature [47–50], and therefore require an open system description of the circuit dynamics. We disregard such process below, which is a good first approximation provided that they occur on slower time scales than the coherent circuit dynamics due to $H(\varphi)$.

## 2.2 Geometric properties of current correlations

Statement (i) inescapably leads to the question of how charge quantization, respectively, how the detector resolution can be measured in a direct and unique fashion. This is no easy feat. Differences in the circuit dynamics (as listed in the introduction) may be of various origins, such that it may be difficult to disentangle charge quantization from other external influences.

Apart from that, it has been discussed in different contexts, that there is a connection between charge quantization and the full-counting statistics of transport [7,38]. Specifically, for quantum circuit theory, full-counting statistics can be formulated along the lines of Ref. [51]. Starting from the von Neumann equation for a given circuit Hamiltonian, $\dot{\rho} = -i[H(\varphi), \rho]$, a counting field $\chi$ [4] is included as a shift in $\varphi$ which is positive (negative) for the forward (backward) propagation, $\rightarrow \dot{\rho}(\chi) = -iH(\varphi + \chi)\rho(\chi) + i\rho(\chi)H(\varphi - \chi)$. The moment generating function $m(\chi)$ can then be obtained via the trace over the density matrix $\rho(\chi)$ [5]. Hence, the moment generating function here inherits its periodicity in $\chi$ from the periodicity of $H$ in $\varphi$.

However, there are two problems. First, we note that the most easily accessible statistical quantities are low cumulants, which are obtained through a Taylor expansion of $m$ in $\chi$ around $\chi = 0$. For instance, the average current and the current noise can be constructed through first and second order derivatives in $\chi$, and thus only probe the local properties of $m$, and not the global properties (i.e., the periodicity). Second, there is the even subtler issue that in leading order, the moment generating function is dominated by the eigenvalues of $H$ only [51], whereas the properties of the eigenbasis, in particular, its $\varphi$-dependence, remain invisible (i.e., they give rise to small corrections). It is however the latter property that we are after, as it contains the information about the detector resolution. On a more figurative level: if we only

---

[4]The counting field $\chi$ is for obvious reasons related to $\varphi$. Note though, that the two objects are nonetheless distinct: one can think of $\chi$ as the "classical" component of $\varphi$, in the sense that it appears with opposite sign for the forward and the backward propagation.

[5]Note that this trace is only trivially 1 for $\chi = 0$. For finite $\chi$, the modified von Neumann equation is not trace-preserving, which is a standard feature in full-counting statistics.

consider the average number of charges transported into a certain region, it does not seem to matter whether or not this charge was measured in discrete packages. With this in mind, it is highly surprising that we are here able to show that there exist certain low-cumulant observables, which *are* sensitive to charge quantization, and can in fact be related to the $\varphi$-dependence of the eigenbasis of $H$ (thus directly probing the $U(\varphi)$ introduced above). As it turns out, while the noise (that is, the current-current correlations) into a particular lead is in leading order not sensitive to charge quantization, the *sum* of auto- and cross-correlations of the supercurrent is. In fact, the leading parts of auto- and cross-correlations mutually cancel, unveiling a purely geometric component.

For this purpose, consider a quantum circuit connected to a left and right contact, with a given phase difference $\varphi$. Two detectors, measuring a current to the left ($I_\text{L}$) and the right ($I_\text{R}$), thus define a finite size region hosting a total charge $2eN$, via the continuity equation, $I_\text{L} + I_\text{R} = 2e\dot{N}$. We stress that while the left and right contacts are well-defined, the same is not guaranteed for the detectors measuring $I_\text{L,R}$, since the detection may occur with a finite spatial resolution (see, e.g., the SQUID detectors in Fig. 1). In accordance with (i), such imperfections can be included as a gauge choice in the Hamiltonian, that is, a gauge where $I_\text{R} = 2e\partial_\varphi H^{(\text{R})}$, and a gauge where $I_\text{L} = -2e\partial_\varphi H^{(\text{L})}$ (the superscript in $H^{(\alpha)}$ indicates the corresponding gauge). Note that for convenience, we defined $I_\text{L}$ in the opposite direction (see arrows above current operators in Fig. 1a).

Let us now show, how the detector properties can be probed by the current auto- and cross-correlations between the left and right contacts, $I_\text{L}$ and $I_\text{R}$. Integrated over a finite measurement time $\tau$, the correlations are individually defined as

$$S_{\alpha\gamma}(\tau) = \frac{1}{\tau} \int_0^\tau dt \int_0^\tau dt' \frac{1}{2} \left\langle \left\{ \delta I_\alpha(t), \delta I_\gamma(t') \right\} \right\rangle, \tag{6}$$

with $\delta I_\alpha = I_\alpha - \langle I_\alpha \rangle$, and $\alpha, \gamma$ are indices for the left and right detectors. Assuming that the system is in the ground state prior to the measurement, we can show for a general quantum circuit (see Appendix A), that the sum of auto- and cross correlations provides

$$\overline{S}_\gamma(\tau) \equiv \sum_\alpha S_{\alpha\gamma}(\tau) = -2\frac{(2e)^2}{\tau}\gamma\text{Im}\left[ \langle 0|_\gamma A\partial_\varphi |0\rangle_\gamma \right], \tag{7}$$

where $A = N\left(1 - |0\rangle_\gamma \langle 0|_\gamma\right)$ is a Hermitian operator, and $|0\rangle_\gamma$ is the ground state of the Hamiltonian in the gauge $H^{(\gamma)}$. When the index $\gamma$ appears as a prefactor, it returns $\pm 1$ for $\gamma = $ R,L, respectively. The above is one of the main results of this work. It tells us that the sum of auto- and cross-correlations $\overline{S}_\gamma$ can be directly connected to a quantity which depends exclusively on the spatial detector resolution, via $N$ and $\partial_\varphi |0\rangle_\gamma$. This establishes statement (ii).

The result for $\overline{S}_\gamma$ in Eq. (7) can be regarded as a transport version of a geometric phase (with the additional $A$), defined in $\varphi$-space instead of the usual $k$-space. In fact, the construction of $\overline{S}_\gamma$ itself was inspired by the measurement of the mean chiral displacement, which has recently been discussed to measure the Zak-Berry phase in SSH chains [52–54]. Being of geometric nature, it should not surprise that the right-hand side of Eq. (7) is gauge-dependent. This is by no means unphysical. In the case of the SSH model, the Zak-Berry phase depends on the choice of the unit cell [55], a gauge choice which is fixed, once a specific chirality measurement is defined [52–54]. In analogy, here the gauge choice relates to the actual spatial detector resolution and thus to charge quantization, and can likewise be regarded as a choice of a "unit cell" in the space of the transported charges.

Furthermore, we note that for a stationary system, the sum of two currents into different contacts should result in a zero expectation value. In our case this is true for large measurement times $\tau \to \infty$ (dc limit). For finite times $\tau$, there remains a finite contribution: while

the system is in its ground state prior to the first measurement (at time $t = 0$), right after the measurement, it will be projected to a nonstationary state, which takes a finite time to decay. Therefore, the finite displacement current, leading to a nonzero $\overline{S}_\gamma$, is purely induced by the projective measurement. Of course, current measurements are most commonly conducted in the pure dc regime, whereas we here propose a more demanding measurement for finite $\tau$. We emphasize however (see also Appendix A) that Eq. (7) is valid in the long $\tau$ limit, i.e., it corresponds to the asymptotic solution for measurement times longer than the time scale of the internal system dynamics given by $H(\varphi)$.

# 3 The Josephson junction array

## 3.1 Array model and detector fuzziness

Let us illustrate the above findings at the example of a JJ array. The $M + 1$ junctions in series form $M$ superconducting islands (see Fig. 1a). Such a circuit is described in terms of lumped elements, with a single charge and phase operator for each island $m$, $N_m$ and $\varphi_m$ ($m = 1, \ldots, M$), satisfying $[\varphi_m, N_{m'}] = i\delta_{mm'}$. The Hamiltonian is given as (see also [42] and references therein),

$$H(\varphi) = \frac{(2e)^2}{2} \sum_{m=1}^{M} \sum_{m'=1}^{M} N_m \left(\mathbf{C}^{-1}\right)_{mm'} N_{m'} - \sum_{m=1}^{M+1} E_J \cos\left(\varphi_m - \varphi_{m-1}\right). \tag{8}$$

The capacitance matrix $[\mathbf{C}]_{mm'} = \left(2C + C_g\right)\delta_{m,m'} - C\delta_{m,m'+1} - C\delta_{m,m'-1}$ gives rise to the charging energies for the coupling of the islands to a common gate, $E_g = (2e)^2/C_g$, and the nearest neighbour interaction, $E_C = (2e)^2/C$.

The ends are connected to two large superconducting contacts, $m = 0, M + 1$, such that both $\varphi_0$ and $\varphi_{M+1}$ are classically well-defined. We apply a total phase difference $\varphi$ across the array (e.g., by closing the contacts to a loop threaded by an external field) which provides a constraint on the phases. Importantly, there remains a choice as to how this constraint is satisfied. These different choices correspond to gauge choices, which are related through a unitary $U(\varphi)$, and can be used to represent the detector properties, as outlined in (i).

We here consider a general detector, coupling to the current $I_m = 2e\partial_{\varphi_m} H$ (at junction $m$) with a certain coupling prefactor $\eta_m$, $I_\eta = \sum_{m=1}^{M+1} \eta_m I_m$ (where we impose $\sum_{m=1}^{M+1} \eta_m = 1$ for normalization). Note that in general, we would have to compute explicitly two sets of $\eta_m$, one for the left detector, $I_L$, and one for the right detector $I_R$. Below, we will however consider a setup where only one of the two detector properties is relevant. At any rate, the detector is in general fuzzy, as it fails to measure a current at a sharply defined device position. The coupling prefactors can be imbedded into the potential energy

$$\overset{\text{detector gauge}}{\rightarrow} -\sum_{m=1}^{M+1} E_J \cos\left(\varphi_m - \varphi_{m-1} + \eta_m \varphi\right), \tag{9}$$

where here, $\varphi_0 = \varphi_{M+1} = 0$. The validity of Eq. (9) can be easily verified, by differentiating the above potential energy with respect to $\varphi$, thus yielding $I_\eta$. Crucially, this Hamiltonian is only $2\pi$-periodic in $\varphi$, when the measurement is spatially sharp ($\eta_{m_0} = 1$ for one $m_0$, and 0 elsewhere).

The coefficients $\eta_m$ have to be computed based on the detector details and the device geometry. For concreteness, inspired by Ref. [56] we here consider a SQUID detector [6]. In Appendix B, we outline a possible single-shot read-out of the currents (via short incoming wave packets, see Fig. 1a) in analogy to standard qubit read-out schemes [57].

As for the JJ array geometry, since we are only interested in a general demonstration of the principle, we use for simplicity a true 1D geometry for the array instead of more complex realistic geometries [43, 44]. The computation of the $\eta_m$ is then accomplished in two steps. First, it requires the Biot-Savart law to compute the magnetic flux piercing the SQUID, $\Phi_S \sim d_S \int dx I(x) / \left[(x - x_S)^2 + d_S^2\right]^{3/2}$ where $x_S$ is the position of the SQUID with respect to the 1D array, and $d_S$ is the distance (see Fig. 1c where the grey area illustrates the Biot-Savart law).

Second, we note that the local current $I(x)$ is not as such represented in terms of the island degrees of freedom, $\varphi_m$ and $N_m$. Based on the lumped element model of the Hamiltonian given in Eq. (8), only the currents at the individual JJs ($I_m$) can be constructed, whereas the Biot-Savart law requires knowledge also of the currents at positions in between two junctions. This issue can be resolved by starting from a more refined model, taking into account the internal degrees of freedom of the islands, and performing a low-frequency approximation, assuming that the detectors cannot resolve correlations on time scales comparable to the internal dynamics (see Appendix C). Based on this, we find that if the position $x$ is in between junctions $m$ and $m+1$, positioned at $x_m$ and $x_{m+1}$ (see Fig. 1c), the local current is given as $I(x) \approx (1 - \delta_m)I_m + \delta_m I_{m+1}$ with $\delta_m = (x - x_m)/(x_{m+1} - x_m)$. When the position of $x$ is at a contact instead of an island, $x > x_{M+1}$ (or $x < x_1$) the current is simply $I(x) \approx I_{M+1}$ ($\approx I_1$), since the contacts are large. Plugging the relationship between $I(x)$ and $I_m$ into the Biot-Savart law, we find $\Phi_S \sim \sum_m \eta_m I_m$, and thus access the coefficients $\eta_m$ (done numerically for the plot in Fig. 1b).

Note that there is a subtlety regarding the detector fuzziness. In principle, for $d_S$ small with respect to the island length scale $\Delta x$ (the distance between two junctions), the detector should be considered sharp on all relevant length scales (since $\Delta x$ is the smallest length scale of the device). However, within the above low-frequency approximation, we see that this does *not* guarantee a sharp measurement of the currents at the junction. Namely, when the detector is placed at $d_S \ll \Delta x$ and in between two junctions $m$ and $m+1$, we find $\eta_m \approx 1 - \delta_m$ and $\eta_{m+1} \approx \delta_m$ and all other $\eta$'s equal to zero. We thus encounter two distinct notions of detector fuzziness. Either the detector is further away from the system than the relevant length scale $d_S > \Delta x$, such that it cannot resolve individual islands. Or the detector is close, $d_S \ll \Delta x$, but situated in between two junctions, $x_m < x_S < x_{m+1}$, such that it fails to resolve the currents of neighbouring junctions $I_m$ and $I_{m+1}$. Both effects are measurable, as we detail below.

## 3.2 Signatures of charge quantization in the superinductor regime

To proceed, we consider the system in the regime $E_C \ll E_J$. Here, the phases $\varphi_m \approx \varphi_m^{(f)} + \delta\varphi_m$ are well localized around the local minima of the potential (Josephson) energy, $\varphi_m^{(f)}$, which are separated by energy barriers. For the general gauge in Eq. (9), we find (see Appendix D.1)

$$\varphi_m^{(f)} = \left(\frac{m}{M+1} - \sum_{m'=1}^{m} \eta_m\right)\varphi + \frac{m}{M+1}2\pi f \mod 2\pi \,. \tag{10}$$

---

[6]The experimental setup in [56] is much more involved than our considerations here, primarily in order to minimize detector backaction. Here, for illustration purposes, we neglect such details. Note furthermore, that we expect that we may in fact not need the same amount of backaction protection as in [56]: as detailed in (ii), the detector-induced projection of the quantum state is actually a wanted side-effect.

Up to the gauge choice, expressed through the prefactors $\eta_m$, this result coincides with Ref. [15]. The integers $f$ represent the distinct local minima positions. Neglecting plasmon excitations [7], we can associate a localized ground state wavefunction to each $f$, which depends in general on $\varphi$ through the position $\varphi_m^{(f)}$ around which it is centered, $|f\rangle_\varphi$. Approximating $H$ in this regime, one receives the low-frequency Hamiltonian (see Appendix D.2)

$$H_{\text{low}}(\varphi) \approx \frac{1}{2}\frac{E_J}{M+1}\sum_f (\varphi + 2\pi f)^2 |f\rangle_\varphi \langle f|_\varphi + E_S \sum_f \Big(|f+1\rangle_\varphi \langle f|_\varphi + |f\rangle_\varphi \langle f+1|_\varphi\Big). \quad (11)$$

Equation (11) is the standard Hamiltonian for a phase slip junction (see, e.g., [58] and references therein), including the energy scale related to quantum phase slips, $E_S$, describing a hopping between different minima $f$. The actual magnitude of $E_S$ is irrelevant for our considerations. For estimates in various regimes we refer to the literature [41, 42, 59].

Importantly, in Eq. (11) we put emphasis on a feature which is often disregarded: the aforementioned $\varphi$-dependence of the basis $|f\rangle$. Without it, a discussion about whether the charge transported across the array is quantized or not, is actually meaningless. Note in particular, when $E_S = 0$, the array acts as a perfect superinductor, which seems to break charge quantization (along the lines of Refs. [12, 16]). However, thanks to the $\varphi$-dependence, it is possible to render $H_{\text{low}}$ $2\pi$-periodic even in this regime. The progression of $\varphi$ by $2\pi$ comes with an increase $f \to f+1$ for the energies $(\varphi + 2\pi f)^2$. If we choose a gauge representing a spatially sharp current measurement of one particular junction current $m_0$ ($\eta_{m_0} = 1$ and $\eta_{m \neq m_0} = 0$), the basis $|f\rangle$ is likewise guaranteed to fulfill $|f\rangle_{\varphi+2\pi} = |f+1\rangle_\varphi$, such that $H_{\text{low}}(\varphi + 2\pi) = H_{\text{low}}(\varphi)$. For any fuzzy measurement, $H$ is no longer $2\pi$-periodic. In particular, there is the opposite extreme to a sharp measurement, where the detector couples to all junction currents equally, $\eta_m \approx 1/(M+1)$. Here, $|f\rangle$ does no longer depend on $\varphi$ at all, $\partial_\varphi |f\rangle = 0$. We refer to this as a "maximally fuzzy" detector, which fails to spatially resolve charges along the entire array.

With this realization, we can thus understand the continuous "quasicharge" first coined in Ref. [12] for linear inductors under a different light. Namely, for $E_S = 0$, the eigenspectrum forms parabolas $\sim (\varphi + 2\pi f)^2$ in $\varphi$-space, shifted by $2\pi$ intervals. The crossings of individual parabolas are protected, and can only be gapped for finite $E_S$. We thus understand that the continuous quasicharge corresponds to the lack of periodicity of the individual eigenvalues (parabolas), whereas charge quantization is independently defined through the *eigenbasis*. Thus, the JJ array in the superinductor limit gives rise to a quantum version of a feature which was recently discussed for sequential electron tunneling [38]: namely that there is a distinction between "effective" charges, defined by means of topological transitions in the eigenspectrum [8] which may well be noninteger, whereas the actual charge quantization is preserved independent of the topology of the eigenspectrum, provided that the charge detection is spatially sharp.

Finally, we show that the information of the $\varphi$-dependence of $|f\rangle$ is accessible through a current-current correlation measurement, according to (ii). Let us position the left current detector sufficiently deep inside the left contact, such that it couples only to $I_{m=1}$. Thus we will see only the properties of the right detector, which can be accessed by $\overline{S}_{\text{R}}$. For $M \gg 1$, we find (see Appendix E)

$$\overline{S}_{\text{R}}(\tau) \approx \overline{S}_{\text{R}}^{(0)} \sum_{m=1}^{M+1} \eta_m^2, \quad (12)$$

where $\overline{S}_{\text{R}}^{(0)} = -(2e)^2 \langle N_m^2\rangle_0 / \tau$ and the local charge fluctuations are $\langle N_m^2\rangle_0 = \sqrt{E_J/E_C}$. Thus,

---

[7]This is justified because we consider the system close to the ground state.

[8]Note that when $E_S$ is exponentially suppressed [41, 59], the absence of a gapping of the eigenspectrum can indeed be rightfully considered as topologically protected.

$\overline{S}_R(\tau)$ depends only on a single system constant (via $\langle N_m^2 \rangle_0$) and otherwise solely on the spatial resolution of the right current detector. Due to $0 < \eta_m < 1$ and $\sum_m \eta_m = 1$, $\overline{S}_R^{(0)}$ is the extremal value of $\overline{S}_R$.

Interpreting $\eta_m$ as the probability to measure the current at junction $m$, the sum $\sum \eta_m^2$ can be related to the Rényi entropy $H_2$ [60], via $\sum_m \eta_m^2 = e^{-H_2}$, characterizing the detector fuzzyness. In Fig. 1b, we show $\overline{S}_R$ as a function of the detector position. In particular, we see that when the resolution is maximal, i.e., the detector can resolve the current locally of a single junction $m_0$ ($\eta_{m_0} = 1$ and $\eta_{m \neq m_0} = 0$) $H_2 \to 0$, such that $\overline{S}_R$ assumes the extremal value $\overline{S}_R^{(0)}$. Any finite fuzziness of the detector will result in a finite entropy, and thus will reduce $\overline{S}_R/\overline{S}_R^{(0)}$. This is generally the case for $d_S > \Delta x$, where $\overline{S}_R/\overline{S}_R^{(0)}$ decreases as the SQUID approaches the array, and couples to an increasing number of junction currents (black curve in Fig. 1b). Let us also note the special case for the maximally fuzzy detector, where $\partial_\varphi |f\rangle = 0$ leads to $\overline{S}_R = 0$. Note however, that the sensitivity is also quite significant for a spatially sharp detector, $d_S \ll \Delta x$, due to the above discussed lumped element effect. Placing the detector in the middle of two junctions, $m_0$ and $m_0 + 1$, we find that $\eta_{m_0} = \eta_{m_0+1} = 1/2$ and all other $\eta_m = 0$. Thus, $\overline{S}_R$ returns only half the extremal value due to the detector being unable to distinguish between two neighbouring junctions, (red curve in Fig. 1b). Overall, we see that the sum of auto- and cross-correlations provides in a very transparent and characteristic way information about the detector properties.

## 4 Conclusion and outlook

We have outlined an intimate relationship between charge quantization and the detector properties via a gauge choice. Furthermore, we have shown that current-current correlations unravel a unique geometric signature which distinguishes between quantized and non-quantized charge measurements. We expect that the above presented results will help in the formulation of quantum circuit theories which appropriately account for charge quantization.

Let us comment on further-reaching repercussions of this work. As indicated in Ref. [23], enforcing charge quantization may render a critical reexamination of circuit theories describing quantum phase slip junctions [37, 58] necessary. The above discussion may provide a first stepping stone to that end. Namely, we expect that the $\varphi$-dependence in $|f\rangle$ leads to an insightful caveat in the well-known exact duality between the Josephson effect and quantum phase slips [58], [both described by the Hamiltonian in Eq. (11), when performing the maps $E_S \leftrightarrow E_J$, $E_J/(M+1) \leftrightarrow E_C$, and $\varphi \leftrightarrow N_g$, where $N_g$ is the offset charge in the JJ]. As elaborated in our work, when measuring charge sharply, the phase-slip Hamiltonian itself should be $2\pi$-periodic in $\varphi$. On the other hand, in a single JJ, at least when likewise measuring integer charges, the junction Hamiltonian is actually not periodic in $N_g$ (the eigenenergies are, but not the *eigenbasis*). Therefore, we expect that the duality should not only include the system parameters, but will have to be extended to a duality of the quantities being measured. The details of this idea shall be developed in subsequent works.

Finally, while it is known that the approximated charge operator in Luttinger liquids does not carry integer charges [5–7], there does to the best of our knowledge not exist a feasible "requantization" procedure beyond ad hoc methods [7]. Importantly, we believe that the formal analogy between Luttinger liquids and the low-energy description of JJ arrays [40, 42] could be exploited to extend the above derived principles from quantum circuit theory to correlated 1D physics. In particular, the highly unexpected fact that low-cumulants of the transport statistics are directly sensitive to charge quantization could be instrumental. Namely it would give an easy theoretical access to predictions of new, "requantized" versions of Luttinger liq-

uid theory, which could likewise be experimentally falsified by means of standard transport measurements.

## Acknowledgements

We acknowledge many interesting and fruitful discussions with Gianluigi Catelani, Thomas Schmidt, and David DiVincenzo.

**Funding information** This work has been funded by the German Federal Ministry of Education and Research within the funding program Photonic Research Germany under the contract number 13N14891.

## A  Derivation of current correlation sum

Starting from the definition of the current correlations, Eq. (6) in the main text, we here show how to arrive at Eq. (7). When taking the sum over the contacts $\alpha$, and having the ground state as the initial state, we may first of all neglect the current expectation values in the definition $\delta I_\alpha = I_\alpha - \langle I_\alpha \rangle$, because $\sum_\alpha \langle I_\alpha \rangle = 0$ in the stationary state, such that,

$$\sum_\alpha S_{\alpha\gamma}(\tau) = \frac{1}{\tau} \int_0^\tau dt \int_0^\tau dt' \text{Re} \left\langle \sum_\alpha I_\alpha(t) I_\gamma(t') \right\rangle_0 , \tag{13}$$

where we in addition used the fact that the expectation value of the anticommutator $\langle \{A, B\} \rangle$ can be written as the real part, $2\text{Re} \langle AB \rangle$. We then express $\int_0^\tau dt \sum_\alpha I_\alpha(t) = 2eN(\tau) - 2eN(0)$, based on the continuity equation in the main text. As a next step, we use the gauge of the Hamiltonian $H^{(\gamma)}(\varphi)$, where $I_\gamma = \gamma 2e \partial_\varphi H^{(\gamma)}$ (as in the main text, when $\gamma$ appears as a factor instead of an index, it takes on the values $\gamma = \pm 1$ for $\gamma = $ R,L). Accordingly, we cast all operators and the initial state into the basis belonging to this gauge, $H^{(\gamma)}(\varphi) |n(\varphi)\rangle_\gamma = \epsilon_n(\varphi) |n(\varphi)\rangle_\gamma$ (where from now on we neglect the explicit $\varphi$-argument for simplicity) to find

$$\sum_\alpha S_{\alpha\gamma}(\tau) = \gamma \frac{2e}{\tau} \int_0^\tau dt \, \text{Re} \left[ \sum_{n \neq 0} e^{i(\epsilon_0 - \epsilon_n)(\tau - t)} \langle 0|_\gamma N |n\rangle_\gamma \langle n|_\gamma I_\gamma |0\rangle_\gamma \right]$$

$$- \gamma \frac{2e}{\tau} \int_0^\tau dt \, \text{Re} \left[ \sum_{n \neq 0} e^{-i(\epsilon_0 - \epsilon_n)t} \langle 0|_\gamma N |n\rangle_\gamma \langle n|_\gamma I_\gamma |0\rangle_\gamma \right], \tag{14}$$

where the sum over $n$ is taken without the ground state, because the $n = 0$ contributions of the first and second lines cancel. Using

$$I_\gamma = 2e \sum_n \left( \partial_\varphi \epsilon_n |n\rangle_\gamma \langle n|_\gamma + \epsilon_n \partial_\varphi |n\rangle_\gamma \langle n|_\gamma + \epsilon_n |n\rangle_\gamma \partial_\varphi \langle n|_\gamma \right) \tag{15}$$

we see that due to the $n \neq 0$ sum in Eq. (14), the $\partial_\varphi \epsilon_n$ part in the current operator does not contribute, since it is diagonal. We are left with

$$\sum_\alpha S_{\alpha\gamma}(\tau) = \gamma \frac{(2e)^2}{\tau} \int_0^\tau dt \, \mathrm{Re} \left[ \sum_{n \neq 0} (\epsilon_0 - \epsilon_n) e^{i(\epsilon_0 - \epsilon_n)(\tau - t)} \langle 0|_\gamma N |n\rangle_\gamma \langle n|_\gamma \partial_\varphi |0\rangle_\gamma \right]$$
$$- \gamma \frac{(2e)^2}{\tau} \int_0^\tau dt \, \mathrm{Re} \left[ \sum_{n \neq 0} (\epsilon_0 - \epsilon_n) e^{-i(\epsilon_0 - \epsilon_n)t} \langle 0|_\gamma N |n\rangle_\gamma \langle n|_\gamma \partial_\varphi |0\rangle_\gamma \right]. \quad (16)$$

Carrying out the time integral, we eventually arrive at

$$\sum_\alpha S_{\alpha\gamma}(\tau) = -2\gamma \frac{(2e)^2}{\tau} \mathrm{Im} \left[ \sum_{n \neq 0} \left[ 1 - e^{i(\epsilon_0 - \epsilon_n)\tau} \right] \langle 0|_\gamma N |n\rangle_\gamma \langle n|_\gamma \partial_\varphi |0\rangle_\gamma \right]. \quad (17)$$

We discard the fast oscillatory contribution $\sim e^{i(\epsilon_0 - \epsilon_n)\tau}$, which is justified for measurement times $\tau > \Delta\epsilon^{-1}$, where $\Delta\epsilon$ is a measure for the level spacing $|\epsilon_n - \epsilon_{n'}|$. Finally, we use the completeness of the basis, $\sum_{n \neq 0} |n\rangle_\gamma \langle n|_\gamma = 1 - |0\rangle_\gamma \langle 0|_\gamma$, to arrive at Eq. (7).

## B  Single-shot projective current measurement

Here, we outline a possible single-shot projective measurement of the current in the JJ array, inspired by standard single-shot readout techniques deployed in superconducting circuits [57].

Overall, the JJ array plus a single SQUID detector, measuring current $I_\alpha$, can be described by a composite system $H \to H + H_{\mathrm{SQUID}}(I_\alpha)$. The SQUID including its contact lines may be modelled as a transmission line, (we stick to a discrete version of the transmission line),

$$H_{\mathrm{SQUID}}(I_\alpha) = \frac{1}{2C_0} \sum_j N_j^2 + \frac{1}{2} \sum_j \frac{1}{L_j(I_\alpha)} \left( \frac{\varphi_{j+1} - \varphi_j}{2e} \right)^2, \quad (18)$$

with $\left[ \varphi_j, N_{j'} \right] = i\delta_{jj'}$. Importantly, the inductance of the transmission line is $L_j = L_0$ for all $j$, except at the SQUID position $j = j_0$, where

$$L_{j_0}^{-1}(I_\alpha) = 8e^2 E_{J,\mathrm{SQUID}} \left[ \cos\left( \frac{\phi_{\mathrm{ext}}}{2} \right) - \lambda \sin\left( \frac{\phi_{\mathrm{ext}}}{2} \right) I_\alpha \right], \quad (19)$$

with the SQUID Josephson energy $E_{J,\mathrm{SQUID}}$, $\phi_{\mathrm{ext}} = 2\pi\Phi_{\mathrm{ext}}/\Phi_0$, where $\Phi_{\mathrm{ext}}$ is an external, tunable flux piercing the SQUID, and the second term is due to the magnetic flux created by the current $I_\alpha$ measured by the SQUID. The factor $\lambda$ is a coupling constant with the units of inverse current. The current coupled to the SQUID is computed along the lines given in the main text.

Suppose a signal in form of a local wave packet is created from one side, and is incoming towards the SQUID (see the wave packets in red and blue in Fig. 1a for an illustration). For simplicity, we imagine that the signal has large amplitudes (such that it can be considered classical) and it should have a short spatial support. Once the signal hits the SQUID, given proportions of the signal will be transmitted and reflected, respectively. The transmitted and reflected portions depend on the value for the current $I_\alpha$, such that the transmitted and reflected amplitudes of the outgoing wave packets (after the scattering at the SQUID) will be entangled with the eigenbasis of $I_\alpha$, thus realizing a projective measurement.

For a successful projective read-out, we provide two figures of merit which we deem central. The first one concerns the influence of the detector when it is idle (that is, in the absence

of a wave packet). Here, equilibrium fluctuations of the SQUID degrees of freedom, $\varphi_j$, will couple to the system via the interaction term $\sim \Delta\varphi_{j_0}^2 I_\alpha$, (with $\Delta\varphi_j = \varphi_{j+1} - \varphi_j$), as it appears in Eqs. (18) and (19). Such fluctuations will lead to stochastic transitions in the JJ array system, between the eigenstates $|n\rangle$ and $|m\rangle$ of $H$, with the corresponding eigenenergies $\epsilon_n$ and $\epsilon_m$. These rates can be computed by standard Fermi's Golden rule

$$\Gamma_{m\to n} = \left[ \lambda E_{J,\text{SQUID}} \sin\left(\frac{\phi_{\text{ext}}}{2}\right) \right]^2 |\langle m| I_\alpha |n\rangle|^2 S_{\Delta\varphi^2}(\epsilon_n - \epsilon_m), \qquad (20)$$

with the SQUID noise spectrum $S_{\Delta\varphi^2}(\omega) = \int dt\, e^{i\omega t} \left\langle \Delta\varphi_{j_0}^2(0) \Delta\varphi_{j_0}^2(t) \right\rangle_{\text{eq}}$, taken with respect to the equilibrium state of the transmission lines. The overlap terms scale as $|\langle m| I_\alpha |n\rangle| \sim 2eE_J$. As long as these rates are slower than the internal JJ array dynamics $\Gamma \ll \Delta\epsilon$ ($\Delta\epsilon \sim |\epsilon_m - \epsilon_n|$ is the characteristic energy scale describing the dynamics of $H$), the description of the JJ array by means of the closed system dynamics $H$ remains valid, even in the presence of the idle SQUID.

Second, in order for the detection process to yield a clean projection onto an eigenstate of $I_\alpha$, the wave packet should be sufficiently short in length, $l_{\text{wave}}$, such that the resulting pulse is short-lived with respect to the closed system dynamics of $H$. This is satisfied for $v_p/l_{\text{wave}} \gg \Delta\epsilon$. The constant $v_p \sim \Delta l/\sqrt{L_0 C_0}$ is the velocity with which the wave packets propagate (where $\Delta l$ is the length scale of the discrete transmission line model, and $\sim 1/\sqrt{L_0 C_0}$ is the plasmon frequency of the transmission line).

## C  Lumped-element current operators

As pointed out in the main text, we are confronted with the problem that the SQUID physically measures a local current $I(x)$ within the JJ array; however, those local degrees of freedom are not represented in the lumped element approach describing the array. In particular, for the lumped elements circuit, there appear only the currents at the Josephson junctions, $I_m$. We show here, how to perform a low-frequency approximation of $I(x)$ in a more refined model, in order to relate it to the operators $I_m$.

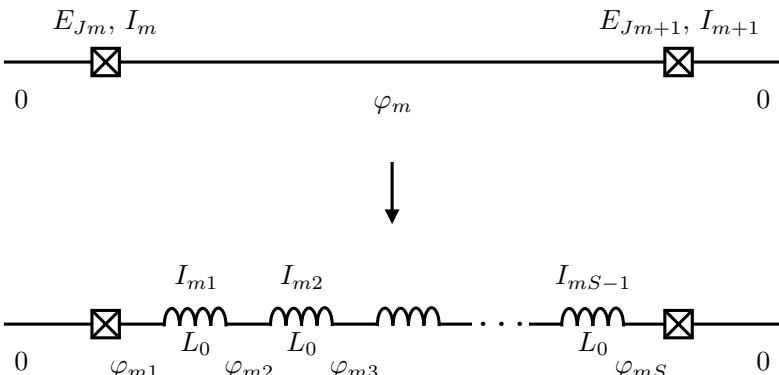

Figure 2: Refining lumped element approach. The island degree of freedom $\varphi_m$ is split into $S$ subparts $\varphi_{ms}$ with $s = 1\ldots S$, which are separated by inductances $L_0$, forming a transmission line. The lumped-element limit is valid when the internal dynamics are very fast, $L_0 \to 0$.

We do this at the example of a single superconducting island, $m$. In order to simplify the problem, we set the neighbouring phases to zero, $\varphi_{m-1} \approx 0$ and $\varphi_{m+1} \approx 0$, see Fig. 2. This

simplification is justified with the foresight, that eventually, for $M \gg 1$, the phase difference across each individual JJ is small (allowing us to extend the argument from $M = 1$ to large $M$). In order to clearly distinguish the junction currents $I_m$ and $I_{m+1}$, we will in this appendix assume different JJ energies, $E_{Jm}$ and $E_{Jm+1}$ for both junctions. Furthermore, we set the phase difference across the contacts to zero. Assuming a true 1D system (see main text), we then add the internal island degrees of freedom, by treating it like a transmission line, that is, taking the island charge and phase, $N_m$ and $\varphi_m$, and partitioning it into $S$ sub islands, such that there are the new degrees of freedom $N_{ms}$ and $\varphi_{ms}$, $[\varphi_{ms}, N_{ms'}] = i\delta_{ss'}$, with $s = 1 \ldots S$, as depicted in Fig. 2. These sub islands are connected through $S-1$ inductances $L_0$. For the sake of simplicity, we here stick to a discrete representation of the internal degrees of freedom. The current at inductance $s$ (for the currents $s = 1, \ldots, S-1$) is $I_s = \vec{I}_s^T \vec{\varphi}_m$ with $(\vec{I}_s)_{s'} = (\delta_{s+1,s'} - \delta_{s,s'})/(2eL_0)$ and $(\vec{\varphi}_m)_{s'} = \varphi_{ms'}$. Approximating the JJs at the end $E_J \cos(\varphi_{m1,mS}) \approx \frac{1}{2}E_J \varphi_{m1,mS}^2 + \text{const.}$, we get the Hamiltonian

$$H_m = \frac{(2e)^2}{2} \sum_{s=1}^{S} \sum_{ss'} N_{ms} \left[\mathbf{C}_m^{-1}\right]_{ss'} N_{ms'} + \frac{1}{2} \sum_{ss'} \frac{\varphi_{ms}}{2e} \left[\mathbf{F}_m\right]_{ss'} \frac{\varphi_{ms'}}{2e}, \tag{21}$$

where the matrices are (as in the main text) denoted with bold font. The matrix for the potential energy reads

$$\left[\mathbf{F}_m\right]_{ss'} = \frac{1}{L_0} \left[(2 - \delta_{s1} - \delta_{sS})\delta_{ss'} - \delta_{ss'-1} - \delta_{ss'+1}\right]$$
$$+ \frac{1}{L_m}\delta_{s1}\delta_{ss'} + \frac{1}{L_{m+1}}\delta_{sS}\delta_{ss'}, \tag{22}$$

having defined the junction inductances as $L_m^{-1} = (2e)^2 E_{Jm}$. The capacitance matrix for the internal degrees of freedom is denoted by $\mathbf{C}_m$. We here consider the lumped element limit $L_0 \to 0$, where one finds that the individual phases of the sub islands all approach a single island value $\varphi_{ms} \approx \varphi_m$, and consequently, $N_{ms} \approx N_m/S$ (the normalization factor $1/S$ appears such that the sum of the sub island charges returns the total island charge $\sum_s N_{ms} = N_m$). In fact, since we consider the limit $L_0 \to 0$, we do not need the specific form of $\mathbf{C}_m$, because the physics is dominated by the structure of $\mathbf{F}_m$ (due to its divergend parts). All we require is that the sum $\sum_{ss'} \left[\mathbf{C}_m\right]_{ss'} = 2C + C_g$, such that it is consistent with the capacitances of the lumped-element model in the main text.

As a matter of fact, when approaching the limit $L_0 \to 0$, all but the lowest mode (i.e., the zero mode) disappear. Hence, we have to compute this leading mode, which is computed by getting the eigenvector with the lowest eigenvalue of the matrix $F_m$, $F_m \vec{v}_0 = f_0 \vec{v}_0$. For $L_{m,m+1} \gg L_0$, we find the eigenvalue $f_0 \approx \left(\frac{1}{L_m} + \frac{1}{L_{m+1}}\right)/S$ and the corresponding eigenvector (including corrections first order in $L_0/L_{m,m+1}$)

$$(\vec{v}_0)_s \approx \frac{1}{\sqrt{S}} + \frac{1}{6}\left[\sqrt{S}\frac{L_0}{L_m} + \frac{1}{2\sqrt{S}}\left(\frac{L_0}{L_m} + \frac{L_0}{L_{m+1}}\right)\right]\left[1 - 3\left(\frac{s}{S} - 1\right)^2\right]$$
$$+ \frac{1}{6}\left[\sqrt{S}\frac{L_0}{L_{m+1}} - \frac{1}{2\sqrt{S}}\left(\frac{L_0}{L_m} + \frac{L_0}{L_{m+1}}\right)\right]\left[1 - 3\left(\frac{s}{S}\right)^2\right]. \tag{23}$$

The current at position $s$, can then be cast into a low-frequency description, by projecting it into the lowest mode only, such that we find

$$I_s = \vec{I}_s^T \vec{\varphi}_m \overset{\text{low-frequency}}{\approx} \vec{I}_s^T \vec{v}_0 \vec{v}_0^T \vec{\varphi}_m, \tag{24}$$

which, for $S \gg 1$, results in

$$I_s \approx \left(1 - \frac{s}{S}\right)\frac{1}{2eL_m}\frac{\sum_{s=1}^{S}\varphi_{ms}}{S} + \left(-\frac{s}{S}\right)\frac{1}{2eL_{m+1}}\frac{\sum_{s=1}^{S}\varphi_{ms}}{S}, \tag{25}$$

which can be expressed in terms of the currents $I_{m,m+1}$ across junction $m, m+1$ in the lumped-element limit

$$I_m = -\frac{1}{2eL_m}\varphi_m \text{ and } I_{m+1} = \frac{1}{2eL_{m+1}}\varphi_m, \tag{26}$$

(note that the minus sign in $I_m$ comes from $\varphi_{m-1} - \varphi_m \approx -\varphi_m$) by realizing that for $L_0 \to 0$, $\frac{\sum_{s=1}^S \varphi_{ms}}{S} \approx \varphi_m$. Going to the continuum limit, we receive

$$I(x) \approx \left(1 - \frac{x - x_m}{x_{m+1} - x_m}\right) I_m + \frac{x - x_m}{x_{m+1} - x_m} I_{m+1}, \tag{27}$$

if the measurement position $x$ is in between the left and right junctions, $m, m+1$, placed at $x_m$ and $x_{m+1}$. This result can be generalized to an array of many junctions, $M > 1$, by realizing that the Cooper pair transport inside the transmission line, via $L_0$, is local. Hence, a similar calculation including more than one island will result in the same linear relationship for each island $m$. This demonstrates the statement in the main text.

As an addendum, let us note that one can use the above result also to discuss the case when the current measurement occurs at a position $x$ inside the large contacts, that is, $x > x_{M+1}$ or $x < x_1$. Note that the JJ array in the main text is closed by a large loop, connecting junctions $m = 1$ and $m = M + 1$, with a size much larger than $\Delta x$. Hence, we may likewise close the manifold $x$ to a loop, and discuss the area between junctions $x_{M+1}$ and $x_1$ in the same fashion as above. For concreteness, let us argue for the case $x > x_{M+1}$. Here, as long as $x$ is close to $x_{M+1}$ compared to the total large loop length, we find $I(x) \approx I_{M+1}$.

# D   Local minima and low-frequency Hamiltonian

## D.1   Local minima

The values $\varphi_m^{(f)}$ are the ones that minimize the potential energy in Eq. (9) in the main text. For illustration, the potential energy landscape is also depicted in Fig. 3. At the local minima, the derivatives $\partial_{\varphi_m}$ of the potential energy have to vanish, hence we find the conditions

$$\sin\left(\varphi_{m+1}^{(f)} - \varphi_m^{(f)} + \eta_{m+1}\varphi\right) - \sin\left(\varphi_m^{(f)} - \varphi_{m-1}^{(f)} + \eta_m\varphi\right) = 0, \tag{28}$$

for $m = 1, \ldots, M$. For small arguments inside the sine functions, we may simplify

$$\varphi_{m+1}^{(f)} - 2\varphi_m^{(f)} + \varphi_{m-1}^{(f)} \approx -(\eta_{m+1} - \eta_m)\varphi \tag{29}$$

For the boundary conditions $\varphi_0 = \varphi_{M+1} = 0$, the solution of Eq. (29) is

$$\varphi_m^{(f)} = \left(\frac{m}{M+1} - \sum_{m'=1}^{m} \eta_{m'}\right)\varphi, \tag{30}$$

as can be seen when plugging Eq. (30) into Eq. (29). The first linear term $\sim m/(M+1)$ ensures that $\varphi_{M+1}^{(f)} = \varphi_{M+1} = 0$, as the boundary condition demands, since, as per definition in the main text, $\sum_{m=1}^{M+1} \eta_m = 1$. While Eq. (30) solves for the approximated condition in Eq. (29), we also have to satisfy the exact conditions in Eq. (28). For this purpose, we need to take into account that there are many solutions, including $\varphi_{M+1} = 2\pi f$, (which is equivalent to $\varphi_{M+1} = 0$, due to the periodicity of the potential energy). These extra solutions are taken into account by

$$\varphi_m^{(f)} = \left(\frac{m}{M+1} - \sum_{m'=1}^{m} \eta_{m'}\right)\varphi + \frac{m}{M+1}2\pi f. \tag{31}$$

Finally, we have to take into account that the potential energy in Eq. (9) is $2\pi$-periodic in each $\varphi_m$, such that the result from Eq. (31) has to be taken up to modulo $2\pi$, resulting in Eq. (10) in the main text. The minima are also shown in Fig. 3a, as red dots.

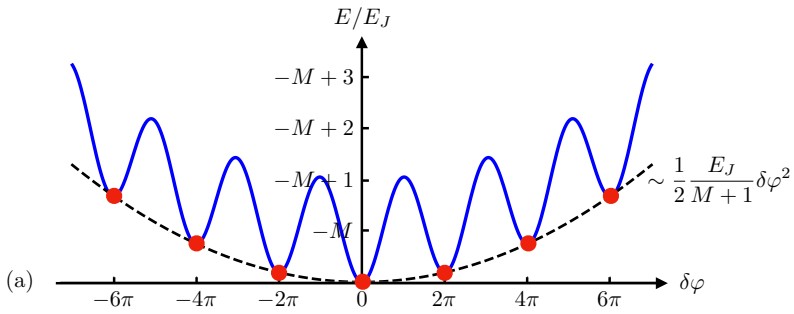

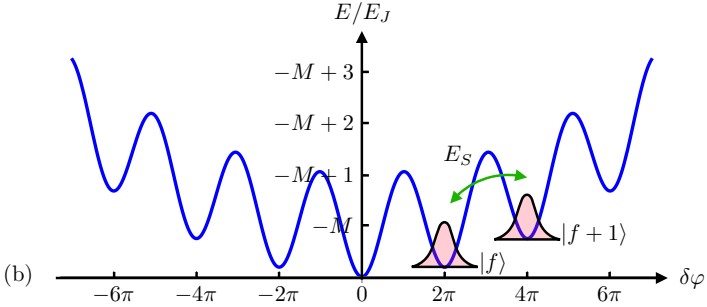

Figure 3: Sketch of the potential energy landscape, i.e., the Josephson energy as defined in Eq. (9), as a function of $\delta\varphi$, where $\varphi_m = \varphi_m^{(f)} + \frac{m}{M+1}\delta\varphi$ for $m = 1, \ldots, M$, and $\varphi_{0,M+1} = 0$. The deviation $\delta\varphi$ is chosen such that for $\delta\varphi$ being multiple integers of $2\pi$, we pass through other minima, that is, for instance $\varphi_m^{(f)} + \frac{m}{M+1}(\delta\varphi + 2\pi) = \varphi_m^{(f+1)} + \frac{m}{M+1}\delta\varphi$. In the plot here, we chose for simplicity $\varphi = 0$, and $M = 105$. In (a) the local minima are denoted as red dots. The energies at the local minima are given by a parabola with respect to $\delta\varphi$ (dashed black line), with the focal length $[2E_J/(M+1)]^{-1}$. In (b) the local wave functions $|f\rangle$, centered around the local minima are depicted. Quantum phase slips, occurring with energy scale $E_S$, are tunneling events between different local wave functions.

## D.2 Low-frequency Hamiltonian

Based on the local minima computed in the section above, one may derive the low-frequency Hamiltonian given in Eq. (11) in the main text, describing the ground state of the JJ array. For this purpose, one expands the full Hamiltonian $H$ given in Eq. (8) locally around the minima, in leading orders of the small deviations $\delta\varphi_m$ from the minimal values, $\varphi_m = \varphi_m^{(f)} + \delta\varphi_m$. Such a procedure is a good approximation for $E_J \gg E_C$, when the minima are separated by high energy barriers. This approximation will be explicitly performed below, in Appendix E, up to quadratic order in $\delta\varphi_m$.

Here, at this stage, the details of this approximation is irrelevant. All that matters is that there is a local Hamiltonian $H_f$ which asymptotically describes the full $H$ close to the local minimum $\varphi_m^{(f)}$, for which however the local potential energy minimum becomes a global one (in essence, a standard tight-binding approach). Thus, within the energy troughs around the minima, we find wave functions (eigenfunctions of $H_f$) which describe local ground and excited states. For the purpose of this work, we are only concerned with the local ground

states, which we denote as $|f\rangle$ (as in the main text) localized around the minima $\varphi_m^{(f)}$, as also schematically depicted in Fig. 3b. Neglecting the excited states (the plasmon excitations, see also Appendix E below), we may use the basis spanned by the localized wave functions $|f\rangle$ to find a low-frequency approximation of the Hamiltonian. The only additional information we need is the energies at the local minima, which depend quadratically on the minima positions, as depicted in the dashed line in Fig. 3a. Consequently, we find,

$$H_{\text{low}} \approx \frac{1}{2}\frac{E_J}{M+1}\sum_f (\varphi + 2\pi f)^2 |f\rangle_\varphi \langle f|_\varphi + \text{const.} \tag{32}$$

As pointed out in the main text, the basis $|f\rangle$ depends on $\varphi$ due to the $\varphi$-dependence of the minima positions $\varphi_m^{(f)}(\varphi)$. In addition, while the different minima are separated by high barriers (in the limit $E_J \gg E_C$), there may still be a quantum tunneling between neighbouring minima, $|f\rangle \longleftrightarrow |f \pm 1\rangle$, the so-called quantum phase slips, as indicated in the main text. Associating energy $E_S$ to these tunneling processes, we arrive at Eq. (11) in the main text.

# E  Relationship between $\overline{S}_\gamma$ and detector resolution

## E.1  Charge correlations

The computation of $\overline{S}_\gamma$ requires the calculation of the charge correlations $\langle N_m N_{m'} \rangle_0$ for the ground state, see Sec. E.2 below. We here compute $\langle N_m N_{m'} \rangle_0$ based on a harmonic approximation of the JJ array Hamiltonian in Eq. (8), valid for $E_J \gg E_C$. That is, for $\varphi_m = \varphi_m^{(f)} + \delta\varphi_m$ for a given $f$, we expand $H$ up to quadratic order in $\delta\varphi_m$,

$$H_f \approx \frac{(2e)^2}{2}\sum_{m=1}^M \sum_{m'=1}^M N_m \left(\mathbf{C}^{-1}\right)_{mm'} N_{m'} + \frac{1}{2}E_J \sum_{m=1}^M \sum_{m'=1}^M \delta\widehat{\varphi}_m (\mathbf{F})_{mm'} \delta\widehat{\varphi}_{m'} + \text{const.}, \tag{33}$$

where the matrix $(\mathbf{F})_{mm'} = 2\delta_{m,m'} - \delta_{m,m'+1} - \delta_{m,m'-1}$ has the exact same shape as the capacitance matrix for $C_g = 0$. In fact, we find that $\mathbf{C} = C\mathbf{F} + C_g \mathbf{1}$, which means that both $\mathbf{C}$ and $\mathbf{F}$ commute and thus have the same eigenvectors. The eigenvectors $\mathbf{F}v_k = f_k v_k$ are standing waves (plasmon excitations) $v_k = \sqrt{2/(M+1)}\sin(km)$, and have the eigenvalues $f_k = 2 - 2\cos(k)$ where $k = \pi l/(M+1)$, $l = 1, 2, 3, \ldots$, is discrete due to the finite size of the system. By means of these eigenvalues and vectors, we may express the charge operator in terms of the $k$-modes as

$$N_m = \sum_k (\vec{v}_k)_m N_k = \sum_k \sqrt{\frac{2}{M+1}}\sin(km)N_k, \tag{34}$$

where $N_k$ can be expressed in terms of the boson operators $\left[b_k, b_{k'}^\dagger\right] = \delta_{kk'}$,

$$N_k = \frac{-i}{\sqrt{2}}\left[\left(\frac{E_J}{E_C}f_k + \frac{E_J}{E_{C_g}}\right)f_k\right]^{\frac{1}{4}}\left(b_k - b_k^\dagger\right). \tag{35}$$

Focussing on the system being in the ground state $b_k |0\rangle = 0$ for all $k$, the charge correlations for the individual islands can be expressed as

$$\langle N_m N_{m'} \rangle_0 = \frac{1}{M+1}\sum_k \sin(km)\sin(km')\sqrt{\left(\frac{E_J}{E_C}f_k + \frac{E_J}{E_{C_g}}\right)f_k}. \tag{36}$$

For $C_g \ll C$, we find

$$\langle N_m N_{m'} \rangle_0 \approx \sqrt{\frac{E_J}{E_C}} \frac{2}{M+1} \sum_l \sin\left(\pi \frac{l}{M+1} m\right) \sin\left(\pi \frac{l}{M+1} m'\right) \left[1 - \cos\left(\pi \frac{l}{M+1}\right)\right], \quad (37)$$

which, after evaluation of the sum over $l$, results in

$$\langle N_m N_{m'} \rangle_0 \approx \langle N_m^2 \rangle_0 \left[\delta_{mm'} - \frac{1}{2}\delta_{mm'+1} - \frac{1}{2}\delta_{mm'-1}\right], \quad (38)$$

with $\langle N_m^2 \rangle_0 = \sqrt{E_J/E_C}$. We observe that the charge correlations extend only over nearest neighbours $m = m' \pm 1$, due to $C_g \ll C$. For larger $C_g$, the correlations would fall off only over longer distances.

## E.2 $\overline{S}_\gamma$ for JJ array

Here, we derive Eq. (12) based on Eqs. (7) and (38). As detailed in the main text, we assume that the left detector is located sufficiently deep inside the left contact, such that $I_L \approx -I_{m=1}$, whereas the right current may be measured in general as $I_R = \sum_m \eta_m I_m$. Consequently, the charge $N$ enclosed by the two interfaces, satisfying $I_L + I_R = 2e\dot{N}$, can be written as

$$N = \sum_m \left(1 - \sum_{m'=1}^{m} \eta_{m'}\right) N_m. \quad (39)$$

Next, we need to compute the $\varphi$-derivative for the ground state. Focussing on $E_J \gg E_C$, we may consider the ground state of the Hamiltonian given in Eq. (11). In the case of $E_S = 0$ (absence of quantum phase slips), the ground state is simply $|0\rangle_\varphi = |f\rangle_\varphi$ for the value of $f$ with the lowest energy $E_J (\varphi + 2\pi f)^2 / (M+1)$. For finite $E_S$, it will be a superposition of different $f$, $|0\rangle_\varphi = \sum_f \beta_f(\varphi) |f\rangle_\varphi$. Therefore, the derivative will in general provide two contributions

$$\partial_\varphi |0\rangle_\varphi = \sum_f \partial_\varphi \beta_f(\varphi) |f\rangle_\varphi + \sum_f \beta_f(\varphi) \partial_\varphi |f\rangle_\varphi. \quad (40)$$

The subsequent calculation is now simplified considerably, due to the overlap between wave functions with different $f$ being exponentially suppressed, and due to $\langle f|N|f\rangle \approx 0$ as well as $\langle f|\partial_\varphi|f\rangle \approx 0$ (in the harmonic approximation, valid for $E_J \gg E_C$). As a consequence, both $\langle 0|\partial_\varphi|0\rangle \approx 0$ and $\langle 0|N|0\rangle \approx 0$, such that we can simplify Eq. (7) to

$$\overline{S}_\gamma(\tau) \approx 2i \frac{(2e)^2}{\tau} \gamma \langle 0|_\gamma N \partial_\varphi |0\rangle_\gamma. \quad (41)$$

Furthermore, for the same reason, the contributions due to $\partial_\varphi \beta_f$ cancel, resulting in

$$\overline{S}_\gamma(\tau) \approx 2i \frac{(2e)^2}{\tau} \gamma \sum_f |\beta_f(\varphi)|^2 \langle f|_\gamma N \partial_\varphi |f\rangle_\varphi. \quad (42)$$

Note now, that $\langle f|_\gamma N \partial_\varphi |f\rangle_\varphi$ is the same for all $f$, such that, due to normalization of the wave function, $\sum_f |\beta_f(\varphi)|^2 = 1$, we eventually find

$$\overline{S}_\gamma(\tau) \approx 2i \frac{(2e)^2}{\tau} \gamma \langle f|_\gamma N \partial_\varphi |f\rangle_\varphi, \quad (43)$$

computed for an arbitrary $f$. We proceed as follows,

$$\partial_\varphi |f\rangle = \sum_m \left( \frac{m}{M+1} - \sum_{m'=1}^{m} \eta_{m'} \right) \partial_m |f\rangle = i \sum_m \left( \frac{m}{M+1} - \sum_{m'=1}^{m} \eta_{m'} \right) N_m |f\rangle , \qquad (44)$$

where for the second identity we used the representation of $N_m$ in $\varphi_m$-space, i.e., $N_m = -i\partial_{\varphi_m}$. The above two result can be plugged into Eq. (43). As a result, we obtain for $\gamma = \mathrm{R}$

$$\overline{S}_{\mathrm{R}} = -\frac{(2e)^2}{\tau} \sum_{m=1}^{M} \sum_{m'=1}^{M} \left( 1 - \sum_{m''=1}^{m} \eta_{m''} \right) \left( \frac{m'}{M+1} - \sum_{m''=1}^{m'} \eta_{m''} \right) \langle N_m N_{m'} \rangle . \qquad (45)$$

Here, we may insert Eq (38). After some algebra, we find

$$\overline{S}_{\mathrm{R}} = -\frac{(2e)^2}{\tau} \langle N_m^2 \rangle_0 \left[ \sum_{m=1}^{M} \left( \eta_m - \frac{1}{M+1} \right)^2 + \left( \frac{M}{M+1} - \sum_{m'=1}^{M} \eta_{m'} \right)^2 + \left( \frac{1}{M+1} - \eta_1 \right) \right] . \qquad (46)$$

Next, we use $\sum_{m'=1}^{M+1} \eta_{m'} = 1$ to simplify the above expression into the form

$$\overline{S}_{\mathrm{R}} = -\frac{(2e)^2}{\tau} \langle N_m^2 \rangle_0 \left[ \sum_{m=1}^{M+1} \left( \eta_m - \frac{1}{M+1} \right)^2 - \left( \eta_1 - \frac{1}{M+1} \right) \right] . \qquad (47)$$

Now, we furthermore use $M \gg 1$ to arrive at

$$\overline{S}_{\mathrm{R}} \approx -\frac{(2e)^2}{\tau} \langle N_m^2 \rangle_0 \left( \sum_{m=1}^{M+1} \eta_m^2 - \eta_1 \right) . \qquad (48)$$

Eventually we consider only cases where the right current $I_{\mathrm{R}}$ is sufficiently different from $-I_{\mathrm{L}}$ (such that the charge $N$ enclosed by the two detectors is finite). In this case, we may assume $\eta_1 \ll 1$, such that we arrive at Eq. (12) in the main text.

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
