# Peer review of "Charge quantization and detector resolution"

_SciPost Physics, doi:SciPost Phys. 10, 093 (2021)_

## Round 3 · Referee Report · Anonymous · 2021-2-3

Strengths

1. The paper addresses an important question of charge quantization in superconducting circuits.

2. The paper is well written.

Weaknesses

1. Certain technical aspects are not clearly explained.

2. The cross correlator of the currents considered in the paper is difficult to measure in experiment. It is only a minor weakness because the focus of paper is theory.

Report

The author presents an interesting study of the interplay between the spatial resolution of a charge detector and the discreetness of the charge transmitted through a superconductor. He finds that only a detector sensitive to the local current may detect individual Cooper pairs.

I find the results presented in the paper very interesting and recommend accepting the paper.

Two typos on pages 16 and 18, where LaTeX references are handled incorrectly, should be fixed.

Requested changes

I recommend optional revision of the text. Namely, I am a bit confused about the physical meaning of the phase \varphi. The author calls it "gauge", but the exact meaning of it remains unclear to me. By looking at Fig. 1a I would assume \varphi = \varphi_{M+1} - \varphi_0, but it is probably incorrect. To me "gauge" implies a global phase factor in the wave function of the whole system having the form \exp[i\varphi]. Alternatively, \varphi could be the counting field, which is also denoted in the text as \chi, or the phase difference across the detector SQUID. It would very useful to discuss this point in more detail.

  • validity: high
  • significance: good
  • originality: high
  • clarity: good
  • formatting: perfect
  • grammar: perfect

Author:  Roman-Pascal Riwar  on 2021-03-24  [id 1329]

(in reply to Report 1 on 2021-02-03)

I warmly thank the referee for the careful reading, the overall very positive review, and the very helpful comments.

The referee asks for clarifications with respect to the significance of \varphi and its relationship to the term “gauge”. In a revised version, modifications at pivotal points will be made, to clearly and unambiguously define both of these terms. Allow me here address the question by the referee, by highlighting three important points.

1) The gauge and the phase \varphi are distinct terms. The gauge choice refers to the choice of the measurement basis, i.e., the basis in which the Hamiltonian is formulated. We can connect different choices (and thus different gauges) by a unitary transformation U. The object \varphi on the other hand is a regular degree of freedom of the circuit. In the old manuscript, the usage of the term “gauge” was only implicitly defined. In the new, revised manuscript, the word gauge, as used throughout the text, will be explicitly defined, in order to avoid confusion.

2) The counting field \chi and the \varphi degree of freedom are related, but ultimately distinct objects, which I will explain in more detail in the revised manuscript. In fact, one can think of the counting field as the classical component of \varphi. This can be seen in the structure of the modified von-Neumann equation describing the full-counting statistics, where \chi appears as a shift in \varphi, which is positive (negative) for the forward (backward) propagation.

3) I think that a further source of confusion was that in the old text, the phase \varphi was merely defined as the variable which is canonically conjugate to the number of charges, that is, the number of Cooper pairs N. However, it was not explicitly specified that the pair of N and \varphi can be interchangeably used to mean either the charge and phase in a given (finite) region, or the charge and phase difference across a given interface. In quantum circuit theory, these two different notions are referred to as node variables, respectively, as branch variables. In the revised manuscript, this point will be clarified right where \varphi is first defined.

Finally, I thank the referee for pointing out the typos on page 16 and 18. These will be fixed in the resubmission.

---

## Round 3 · Referee Report · Anonymous · 2021-3-9

Strengths

1. The paper appears scientifically sound and is reasonably well written.
2. The finding with a Berry-type phase appearing in the current correlators, characterizing the spatial resolution pf the current detector is new and interesting.
3. The example with a Josephson junction array is helpful and pedagogical.

Weaknesses

1. The suggested quantity to measure experimentally is a short time current correlator, going to zero in the long time limit, typically the regime considered in experiments.

Report

In my opinion the criteria for publication is met. It is a solid paper, new physics is presented, it opens up for interesting further work in the field and it provides a possible way to address the lingering questions concerning quantization of charge (or not) in quantum transport.

Requested changes

No particular suggestions for changes

  • validity: high
  • significance: good
  • originality: good
  • clarity: good
  • formatting: good
  • grammar: excellent

Author:  Roman-Pascal Riwar  on 2021-03-24  [id 1330]

(in reply to Report 2 on 2021-03-09)

I warmly thank the referee for his careful reading of the manuscript and the favourable report.

While I am happy to learn that the referee did not find any particular need for changes, I would nonetheless like to briefly comment on what the referee considers the weakness of the paper:

I fully agree with the referee that the most critical point with respect to an experimental verification is the fact that current correlations at finite times need to be measured. Note however that the result presented in the paper is the LONG-TIME asymptotic of the correlations. Hence, I expect that the hurdle on the experimental side may not be as high, since the quantity is (in terms of time scales) not far away from a pure dc noise measurement, and may still be accessible via low-frequency techniques. A comment to that end will be added in the revised manuscript.

---

## Round 4 · Author Response

Dear editor,

Thank you for providing the referee reports. I am very happy to learn that both of the reports are favourable and recommend the publication of the manuscript. One of the referees suggested optional changes, which have been fully addressed in the new revised manuscript (see list of changes, in particular changes 2-5). I herewith resubmit this new manuscript for you consideration.

Sincerely yours,
the author.

---

## Round 4 · List of Changes

1) On page 4, a footnote is added to comment on the completeness of the charge eigenbasis defined in Eq. (2).

2) On page 5, in the paragraph beginning with “The above leads us to our first observation …”, the term "gauge choice" is explicitly defined, in that it refers to the different choices for the detector basis.

3) On page 5, in the paragraph beginning with “In the here considered context …”, a phrase and a new reference ([46] Burkard et al., 2004) is added to highlight that within the paper, the circuit degrees of freedom N and \varphi are interchangeably used to mean either the charge and phase inside a given region (node variables), or the charge and phase difference across an interface (branch variables).

4) On page 7, a footnote is added to explain the nature of the counting field \chi, in particular pointing out that \chi can be regarded as the classical component of \varphi.

5) On page 10, the phrase following Eq. (10), is slightly modified, in order to clearly refer to the above defined term "gauge choice".

6) On page 8, in the final paragraph of Sec. 2.2, a phrase is added to stress that the central result, the current correlator in Eq. (7), is valid in the limit for long measurement times, in particular, measurement times longer than the internal circuit dynamics.

7) The name of a colleage is added in the list of people in the acknowledgements.

---

## Editorial Decision

published